# Determinants of underweight and overweight/obesity among people with tuberculosis in Kampala, Uganda: A cross-sectional study

Simon Kyazze[1], Saidi Appeli[2], Joseph Baruch Baluku[3,4], Jonathan Izudi[1,5,6]*

1 Institute of Public Health and Management, Clarke International University, Kampala, Uganda, 2 Department of Agribusiness and Extension, Faculty of Agriculture and Animal Sciences, Busitema University, Soroti, Uganda, 3 Division of Pulmonology, Kiruddu National Referral Hospital, Kampala, Uganda, 4 TB Research Group, Makerere University Lung Institute, Kampala, Uganda, 5 Department of Community Health, Mbarara University of Science and Technology, Mbarara, Uganda, 6 Directorate of Graduate Training, Research, and Innovation, Muni University, Arua, Uganda

* jonahzd@gmail.com

## Abstract

### Background

Malnutrition significantly contributes to mortality among people with tuberculosis (TB). However, evidence on factors associated with the specific forms of malnutrition, specifically underweight and overweight/obesity, beyond clinical determinants, remains limited in many settings. We investigated the prevalence and determinants of underweight and overweight/obesity among people with pulmonary TB across five health facilities in Kampala, Uganda.

### Methods

This analytic cross-sectional study involved people with pulmonary TB, either clinically diagnosed or bacteriologically confirmed, aged ≥18 years sampled across five health facilities in Kampala, Uganda. Nutritional status was assessed using body mass index (BMI, kg/m²) and categorized as underweight (<18.5), normal weight (18.5–24.9), and overweight/obese (≥25.0). To identify factors independently associated with nutritional status, normal weight was considered as the reference category in a multinomial logistic regression analysis, adjusting for multiple covariates and clustering at the health facility level. The measure of association was the adjusted relative risk ratios (aRRR) and the corresponding 95% confidence intervals (CI).

### Results

Of the 818 participants studied, 417 (51.0%) had normal weight, 302 (36.9%) were underweight, and 99 (12.1%) were overweight or obese. Adjusted analysis showed that being underweight was associated with household food insecurity (aRRR 2.04,

**Data availability statement:** All relevant data are within the paper and its Supporting information files.

**Funding:** The author(s) received no specific funding for this work.

**Competing interests:** The authors have declared that no competing interests exist.

95% CI: 1.48–2.80) while being overweight or obese was associated with self-employment (aRRR 2.26, 95% CI: 1.35–3.79) and being newly diagnosed with TB (aRRR 2.10, 95% CI: 1.30–3.41).

## Conclusion

This study, conducted among people with pulmonary TB in an urban setting in Uganda, showed that underweight and overweight/obesity are prevalent. Furthermore, the study showed that food insecurity is associated with being underweight, while being overweight or obese is associated with being self-employed or newly diagnosed with TB. Therefore, TB control programs need to regularly assess the nutritional status of people with TB to mitigate the effect of being underweight or overweight on treatment outcomes.

## Introduction

Malnutrition is prevalent among people with tuberculosis (TB), and it negatively impacts treatment outcomes [1–3]. For instance, people with TB who are malnourished are more likely to experience delayed sputum smear conversion, reduced treatment success, and higher mortality when compared with those who are well-nourished [4,5]. Furthermore, malnutrition, particularly underweight, contributes to higher mortality rates among people with TB [6]. The rising prevalence of both underweight and overweight/obesity among people with TB further compounds mortality. Therefore, understanding the full scope of malnutrition—underweight and overweight or obesity—is important for improving treatment outcomes among people with TB.

In low-income countries such as Uganda, TB and malnutrition are common. A systematic review and meta-analysis that involved 48,598 people with TB reported a 48.0% pooled prevalence of malnutrition, with factors such as testing positive for *Mycobacterium TB*, low income, and rural residence being strongly associated with malnutrition [7]. Observational epidemiological studies further show that undernutrition, particularly underweight, is prevalent among people with TB and that it reduces treatment success and increases mortality [5,8]. A case-control study found that household contacts of people with multi-drug-resistant TB who are malnourished are more likely to develop TB when compared with household contacts of those with multi-drug-resistant TB who are well-nourished [9], highlighting the broader public health implications of malnutrition. Additionally, malnourished household contacts are more likely to get infected with TB than well-nourished household contacts [10]. This highlights the importance of nutrition supplementation, not just to people with TB, but also to their household contacts. However, the majority of the existing research evidence around malnutrition among people with TB has focused predominantly on undernutrition and the associated factors, with few studies investigating its effect on treatment outcomes.

While researching undernutrition among people with TB is important, previous studies have not focused on the growing burden of overweight and obesity, which

also significantly contributes to the burden of malnutrition. Furthermore, the factors that influence both underweight and overweight or obesity among people with TB, beyond clinical determinants, remain inadequately understood. Moreover, the majority of past studies have combined both underweight and overweight or obesity as a single category of malnutrition.

This masks the meaningful differences between these conditions regarding the independent determinants and potential impact on people with TB. Accordingly, we investigated the prevalence and determinants of both underweight and overweight/obesity among people with clinically diagnosed and bacteriologically confirmed pulmonary TB across five health facilities in Kampala, Uganda. This evidence may contribute to informing targeted and effective interventions to improve the nutritional status of people with TB, which, in turn, may enhance treatment outcomes.

## Methods and materials

### Study design and setting

This analytic cross-sectional study was conducted at five health facilities in Kampala, Uganda. The study sites included Kisenyi Health Centre (HC) IV, Kawaala HC III, Komamboga HC III, Kitebi HC III, and Kisugu Health Centre III. These health facilities provide TB care per national TB treatment guidelines and have been previously described [11–13]. This study adhered to the Strengthening the Reporting of Observational Studies in Epidemiology (STROBE) guidelines for cross-sectional studies [14].

### Study population and sampling

The study population consisted of individuals aged 18 years or older with pulmonary TB, either clinically diagnosed or bacteriologically confirmed. Individuals with bacteriologically confirmed pulmonary TB were those with a biological specimen that was positive for *Mycobacterium TB* on smear microscopy, culture, or molecular tests like GeneXpert. The participants were treated with the standard six-month anti-TB regimen that comprised rifampicin (R), isoniazid (H), pyrazinamide (Z), and ethambutol (E) for two months, followed by rifampicin and isoniazid for four months (2RHZE/4RH). Participants were consecutively sampled across the study sites until the required sample size was achieved.

### Study variables and data collection

The outcome variable was nutritional status, computed using the body mass index (BMI) as weight (kilograms, kg) divided by height (meters squared, m²). BMI was categorized as underweight (BMI < 18.5 kg/m²), normal weight (BMI 18.5–24.9 kg/m²), or overweight/obese (BMI ≥ 25.0 kg/m²). The independent variables included sociodemographic, clinical, dietary, and socio-behavioral factors.

Sociodemographic factors included age (categorized as <25 vs. ≥ 25 years), sex (male or female), marital status (married or never married), educational level (none/primary vs. secondary and above), and employment status (employed vs. unemployed). Clinical factors included human immunodeficiency virus (HIV) status, TB treatment history, such as new TB diagnosis or retreatment TB, anti-TB regimen type, and mental well-being. Sociobehavioral factors included TB-related stigma, smoking, alcohol consumption, and household violence. Dietary factors included food insecurity and dietary diversity. Alcohol consumption was measured using the Alcohol Use Disorders Identification Test–Consumption (AUDIT-C), and the scores ranged from 0 to 12. Food insecurity (FI) was measured using the Food Insecurity Experience Scale (FIES), a global reference scale developed by the Food and Agriculture Organization (FAO). The FIES consists of nine items, each scored as 1 (affirmative) or 0 (non-affirmative), yielding a total score of 0–8. Participants scoring 0–3 were classified as food secure, while those scoring 4–8 were considered food insecure. Dietary diversity was assessed using the World Food Programme (WFP) Food Consumption Score (FCS), based on a 24-hour dietary recall. Scores of ≤21 indicated poor dietary diversity, 21.5–35 borderline dietary diversity, and >35 acceptable dietary diversity. In this study, inadequate

dietary diversity included poor and borderline categories, while adequate dietary diversity referred to the acceptable level category. The entire data was collected between September 02, 2024, and November 29, 2024. Five trained and supervised research assistants collected the data using a standardized questionnaire. Each research assistant held at least a diploma in health sciences and a minimum of three years of quantitative data collection experience. All the data collection took place within the health facility premises after obtaining informed consent.

## Quality control measures

To ensure data validity and reliability, the questionnaire was pre-tested in a non-study area before the data collection. Also, the research assistants received training on the study protocol, informed consent procedures, ethical research conduct, and data collection techniques. Quality control measures, including logical skips, range restrictions, and alerts, were implemented during data entry to ensure data integrity. Filled-out questionnaires were reviewed for completeness by data entrants before the data entry, and random checks were also implemented to ensure the data entered was accurate.

## Statistical methods

**Sample size determination.** This study aimed to determine the prevalence of malnutrition (objective 1) and the associated factors (objective 2). Therefore, the sample size calculation followed the confidence interval approach using a single proportion formula for objective 1 and the statistical significance approach using a two-proportion formula for objective 2, as shown in previous studies [15,16]. First, based on a reported malnutrition prevalence of 46% among adults with TB at a national referral hospital (the Mulago National Referral Hospital) TB clinic [17], the Kish and Leslie formula estimated that 382 participants were needed, assuming a 95% confidence level and a 5% margin of error. Adjusting for a 2.6% non-response rate, the final sample size was 393. Second, using the statistical significance approach, we hypothesized that people with TB who smoke have a higher prevalence of malnutrition compared to those who do not smoke. In Nambi's study, the prevalence of malnutrition among smokers was 53.6%. To detect a 10% difference in malnutrition between smokers and non-smokers at a 5% level of significance ($\alpha = 0.05$) and 80% statistical power, we calculated a required sample size of 822 participants using a two-sided Chi-square test. Overall, we considered the larger sample size of 822 participants because it allowed us to adequately address both objectives.

**Data analysis.** The analysis was performed in R version 4.0.2. We used descriptive statistics to summarize categorical variables (for example, sex and employment status) as frequencies and percentages. We cross-tabulated the independent variables by nutritional status (normal weight, underweight, and overweight/obesity) and assessed differences in their distribution using tests of statistical significance. The Chi-square test was used to assess whether or not there was a significant difference among the levels of the dependent variable. In the inferential level of analysis, normal weight (coded as "0") served as the reference category, while underweight (coded as "1") and overweight/obesity (coded as "2") were the comparison groups. This coding scheme defined the outcome as a nominal variable. Besides, some types of models, including the multinomial logistic regression model, also known as the baseline-category logit model, may be used whether the response variable is ordinal or nominal. We ran a multinominal logistic regression model to identify factors associated with underweight and overweight/obesity, with normal weight as the reference category, adjusting for multiple covariates and clustering at the health facility level.

The model covariates were selected *a priori* based on existing literature and their social and biological relevance. Findings were reported as adjusted relative risk ratios (aRRR) with the respective 95% confidence intervals (CI). We evaluated the model fit using McFadden's pseudo-$R^2$ value, which ranges from 0 to just under 1. McFadden's values below 0.10 indicate poor fit, 0.10 to 0.19 suggest modest fit, 0.20 to 0.39 indicate good fit, and values of 0.40 or higher reflect excellent fit. Additionally, we tested the independence of irrelevant alternatives (IIA) assumption, one of the most important assumptions in a multinominal logit model, using the Hausman test. This assumption suggests that the odds of choosing between any two alternatives are independent of the presence or characteristics of other alternatives.

**Baseline-category logit model:** The baseline-category logit model is generally described as:

$$\log_e\left[\frac{\pi\left(Y = j \mid x_1, x_2, \ldots, x_J\right)}{\pi\left(Y = J \mid x_1, x_2, \ldots, x_J\right)}\right] = \beta_{0j} + \beta_{1j}x_1 + \beta_{2j}x_2 + \ldots + \beta_{pj}\cdots, j = 1, 2, \ldots, J - 1$$

Where;

J = Number of categories of Y (Nutritional status)

Y = Multicategory response variable (Nutritional status)

$\pi$ = Response probabilities

$\beta_{0j}$ = Intercepts

$x_1, x_2 \ldots x_p$ = Covariates for the respective case

$\beta_{1j}, \beta_{2j} \ldots \beta_{pj}$ = Regression coefficients

## Ethical issues

The study was approved by the Clarke International University Research Ethics Committee (CIU-REC, CLARKE-2024–1100, dated July 23, 2024). Administrative clearance was obtained from the Kampala Capital City Authority Directorate of Public Health and Environment (DPHE/KCCA/1301/01, dated August 13, 2024). All participants gave written informed consent after receiving detailed explanations about the study procedures, potential benefits, risks, time compensation, and voluntary participation, including the right to withdraw at any stage. Confidentiality was maintained through anonymized data collection and secure data storage.

## Results

### Characteristics of participants

Table 1 presents the demographic characteristics of the participants. We studied 818 participants (818/822, a 99.5% response rate), the majority of whom were from Kisenyi HC IV (24.7%), aged 25 years or older (62.6%), male (52.9%), married, self-employed like engaged in small-scale business, had attained at least a secondary level of education, had low socioeconomic status, and were newly diagnosed with pulmonary TB. Additionally, many participants (91.1%) had HIV, came from food-insecure households (58.1%), never smoked cigarettes (86.7%), and experienced household violence (73.2%). We found statistically significant differences in nutritional status by type of employment, socioeconomic status, TB treatment history, and household food insecurity.

### Nutritional status of the participants

Among the 818 participants, 417 (51.0%) had normal weight, 302 (36.9%) were underweight, and 99 (12.1%) were overweight or obese (Fig 1).

### Factors associated with underweight and overweight/obesity among people with TB

After adjusting for all important factors in the multinominal logistic regression analysis, the findings in Table 2 showed that being underweight rather than having a normal weight was more likely among participants from food-insecure households compared to food-secure households (adjusted relative risk ratio [aRRR] 2.04, 95% CI: 1.48–2.80). Conversely, overweight or obesity was more likely among participants who were self-employed than those without any form of employment (aRR 2.26, 95% CI: 1.35–3.79) and among individuals newly diagnosed with TB than those being retreated for TB (aRRR 2.10, 95% CI: 1.30–3.41). The remaining factors were not significantly associated with underweight or overweight/obesity. McFadden's pseudo $R^2$ for the final multinomial logistic regression model was 0.1, indicating a modest model fit. While pseudo-$R^2$ values in logistic models are generally lower than those in linear models, a value of 0.1 is considered

**Table 1. Characteristics of participants.**

| Variables | Level | All (n = 818) | Normal weight (n = 417) | Underweight (n = 302) | Overweight/ obesity (n = 99) | P-value |
|---|---|---|---|---|---|---|
| Health facility | Kisenyi Health Centre IV | 202 (24.7) | 108 (25.9) | 72 (23.8) | 22 (22.2) | 0.915 |
| | Kawaala HC III | 180 (22.0) | 94 (22.5) | 65 (21.5) | 21 (21.2) | |
| | Komamboga HC III | 180 (22.0) | 91 (21.8) | 67 (22.2) | 22 (22.2) | |
| | Kitebi HC III | 180 (22.0) | 92 (22.1) | 66 (21.9) | 22 (22.2) | |
| | Kisugu HC III | 76 (9.3) | 32 (7.7) | 32 (10.6) | 12 (12.1) | |
| Age categories (years) | <25 | 306 (37.4) | 167 (40.0) | 102 (33.8) | 37 (37.4) | 0.229 |
| | 25 and above | 512 (62.6) | 250 (60.0) | 200 (66.2) | 62 (62.6) | |
| Sex | Male | 433 (52.9) | 219 (52.5) | 168 (55.6) | 46 (46.5) | 0.276 |
| | Female | 385 (47.1) | 198 (47.5) | 134 (44.4) | 53 (53.5) | |
| Marital status | Married | 562 (68.7) | 285 (68.3) | 218 (72.2) | 59 (59.6) | 0.062 |
| | Not married | 256 (31.3) | 132 (31.7) | 84 (27.8) | 40 (40.4) | |
| Type of employment | Nothing/ none | 312 (38.1) | 177 (42.4) | 110 (36.4) | 25 (25.3) | 0.005 |
| | Self-employed | 506 (61.9) | 240 (57.6) | 192 (63.6) | 74 (74.7) | |
| Level of education | Primary or None | 324 (39.6) | 161 (38.6) | 117 (38.7) | 46 (46.5) | 0.330 |
| | Secondary and above | 494 (60.4) | 256 (61.4) | 185 (61.3) | 53 (53.5) | |
| Socioeconomic status | Low | 283 (34.6) | 136 (32.6) | 98 (32.5) | 49 (49.5) | 0.025 |
| | Moderate | 280 (34.2) | 147 (35.3) | 106 (35.1) | 27 (27.3) | |
| | High | 255 (31.2) | 134 (32.1) | 98 (32.5) | 23 (23.2) | |
| TB treatment history | New | 435 (53.2) | 219 (52.5) | 147 (48.7) | 69 (69.7) | 0.001 |
| | Retreatment | 383 (46.8) | 198 (47.5) | 155 (51.3) | 30 (30.3) | |
| HIV serostatus | Negative | 73 (8.9) | 38 (9.1) | 28 (9.3) | 7 (7.1) | 0.786 |
| | Positive | 745 (91.1) | 379 (90.9) | 274 (90.7) | 92 (92.9) | |
| Dietary diversity | Low | 568 (69.4) | 296 (71.0) | 204 (67.5) | 68 (68.7) | 0.606 |
| | High | 250 (30.6) | 121 (29.0) | 98 (32.5) | 31 (31.3) | |
| Household food insecurity | No | 343 (41.9) | 193 (46.3) | 93 (30.8) | 57 (57.6) | <0.001 |
| | Yes | 475 (58.1) | 224 (53.7) | 209 (69.2) | 42 (42.4) | |
| Alcohol consumption | Low risk | 310 (37.9) | 142 (34.1) | 123 (40.7) | 45 (45.5) | 0.061 |
| | Increasing risk | 355 (43.4) | 203 (48.7) | 119 (39.4) | 33 (33.3) | |
| | High risk | 94 (11.5) | 41 (9.8) | 39 (12.9) | 14 (14.1) | |
| | Possible dependence | 59 (7.2) | 31 (7.4) | 21 (7.0) | 7 (7.1) | |
| Tobacco smoking | No | 709 (86.7) | 357 (85.6) | 259 (85.8) | 93 (93.9) | 0.076 |
| | Yes | 109 (13.3) | 60 (14.4) | 43 (14.2) | 6 (6.1) | |
| Experienced household violence | No | 219 (26.8) | 109 (26.1) | 78 (25.8) | 32 (32.3) | 0.411 |
| | Yes | 599 (73.2) | 308 (73.9) | 224 (74.2) | 67 (67.7) | |

acceptable in epidemiological research, particularly since the outcome is influenced by multiple unmeasured factors. Also, the results showed that the independence of irrelevant alternatives assumption (IIA) was not violated (p-value = 0.165).

## Discussion

We studied the prevalence of underweight and overweight/obesity and the associated factors among people with clinically diagnosed and bacteriologically confirmed pulmonary TB in Kampala, Uganda. We found that 13.2% of the participants have a normal weight, 36.9% are underweight, and 12.1% are overweight or obese. This finding is consistent with the

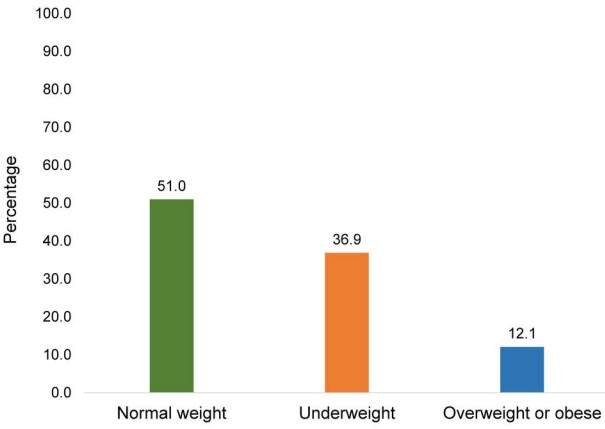

**Fig 1. Bar graph showing nutritional status among people with TB.**

8.7% prevalence of overweight but lower than the 43.1% prevalence of normal weight and underweight (48.2%) among people with TB in a past study in western Ethiopia [18]. The difference in the normal and underweight prevalence rates between the present and past studies may be explained by varying sociodemographic, economic, and dietary factors across the study settings.

This study showed that food insecurity is associated with being underweight, while being self-employed and newly diagnosed with pulmonary TB is associated with being overweight or obese. The finding that food insecurity is associated with being underweight is consistent with previous studies. A case-control study among people with TB in Brazil reported a 26.2% prevalence of underweight, with 64% of the participants reporting some degree of food insecurity [19]. This finding suggests a potential link between food insecurity and underweight, which is consistent with our results. Evidence from epidemiological studies conducted in other populations supports our findings as well. For example, in India, a study found that 27% of adults aged ≥60 years were underweight, and food-insecure individuals were significantly more likely to be underweight than their food-secure counterparts [20]. Similarly, in households with a child or children with HIV, the prevalence of food insecurity was 59%, with more stunted children among food-insecure than in food-secure households [21]. The study's findings may be explained by limited access to nutrient-rich foods among people with TB from food-insecure households, resulting in chronic energy and protein deficiencies, hence contributing to underweight. Besides, in food-insecure households, meal skipping and reduced portion sizes are the most frequent coping mechanisms, leading to prolonged fasting, muscle breakdown, and eventually weight loss. Furthermore, food-insecure households are characterized by inadequate dietary diversity, leading to restricted micronutrient intake, hence exacerbating underweight.

We found that people with newly diagnosed TB are more likely to be overweight or obese compared to those with retreatment TB. This is a unique finding as there is limited literature to support it. However, following extensive literature searching, we identified a study that reported that people with retreatment TB have a reduced likelihood of weight gain compared to those with newly diagnosed TB [22]. Our findings may be explained by the prolonged catabolic effects of TB infection in people with retreatment TB. TB is a known hypermetabolic disease that induces increased energy expenditure, loss of appetite, and muscle wasting. These changes contribute to weight loss. Therefore, people with retreatment TB may experience persistent and increased energy expenditure, appetite loss, and muscle wasting compared to those newly diagnosed with TB. This leads to prolonged undernutrition and difficulty in weight gain among people with retreatment TB when compared with those newly diagnosed with TB.

**Table 2. Results of a multinominal logistic regression analysis of factors associated with underweight and overweight or obesity among people with TB.**

| Variables | Levels | Underweight vs. normal weight | | Overweight/ obesity vs. normal weight | |
|---|---|---|---|---|---|
| | | aRRR | 95% CI | aRRR | 95% CI |
| Age categories years | <25 | 1 | | | |
| | 25 and above | 1.32 | 0.95-1.82 | 0.86 | 0.53-1.41 |
| Sex | Male | 1 | | 1 | |
| | Female | 0.91 | 0.67-1.24 | 1.27 | 0.80-2.01 |
| Level of education | Primary or none | 1 | | | |
| | Secondary and above | 1.00 | 0.73-1.37 | 0.79 | 0.50-1.25 |
| Type of employment | Nothing/ none | 1 | | 1 | |
| | Self-employed | 1.32 | 0.96-1.82 | **2.26**** | **1.35-3.79** |
| Marital status | Married | 1 | | 1 | |
| | Not married | 0.82 | 0.58-1.15 | 1.53 | 0.95-2.47 |
| Socioeconomic status | Low | 1 | | 1 | |
| | Moderate | 1.01 | 0.69-1.47 | 0.58 | 0.33-1.01 |
| | High | 1.02 | 0.69-1.51 | 0.57 | 0.32-1.03 |
| Household food insecurity | No | 1 | | 1 | |
| | Yes | **2.04***** | **1.48-2.80** | 0.69 | 0.43-1.10 |
| Alcohol consumption scores | 1-unit increase | 0.95 | 0.79-1.13 | 0.90 | 0.69-1.18 |
| Tobacco smoking | No | 1 | | 1 | |
| | Yes | 1.01 | 0.65-1.56 | 0.43 | 0.18-1.04 |
| TB treatment history | Retreatment | 1 | | 1 | |
| | New | 0.85 | 0.63-1.15 | **2.10**** | **1.30-3.41** |
| HIV serostatus | Negative | 1 | | 1 | |
| | Positive | 0.92 | 0.54-1.57 | 0.95 | 0.39-2.28 |
| Dietary diversity | Inadequate | 1 | | 1 | |
| | Adequate | 1.15 | 0.82-1.61 | 1.10 | 0.66-1.83 |

**Note:** 1) aRRR: Adjusted Relative Risk Ratio; 2) Exponentiated coefficients are for adjusted relative risk ratio (aRRR); 95% confidence intervals in brackets; and 3) Statistical significance codes at 95% CI: *$p<0.05$, **$p<0.01$, ***$p<0.001$; 4) All analysis adjusted for clustering effect at health facility level. Bolded figures are statistically significant findings at a 5% significance level.

The study showed that people with TB who are self-employed (for example, engaged in small-scale businesses) are more likely to be overweight or obese compared to their unemployed counterparts. This finding may be explained by several interconnected factors.

First, self-employment often serves as a proxy for relatively higher socioeconomic status, which is typically associated with greater access to food, including energy-dense and processed foods.

These factors collectively contribute to increased caloric intake, hence, overweight or obesity. Additionally, individuals with high socioeconomic status can afford conveniences that reduce physical activity, such as motorized transport. This may contribute to reduced energy expenditure, hence overweight or obesity. Second, self-employed people with TB may exhibit dietary patterns characterized by higher consumption of fast or processed foods, which are typically rich in fats, sugars, and refined carbohydrates. These dietary behaviors are strongly linked to weight gain and obesity. Overall, the combination of low physical activity and higher energy intake creates a positive energy balance, thereby increasing the likelihood of overweight and obesity among the subgroup. Appiah and colleagues showed that employment, whether formal or informal, is associated with malnutrition measured as underweight and overweight/obesity combined [23].

Furthermore, in a previous study, a higher educational level, an indicator of employment, was associated with increasing BMI (potentially overweight or obesity) among people with TB [24].

## Implications of study findings

The implications of our findings include the need for integrated public health interventions, including nutritional supplementation, social protection programs such as cash transfers and food aid, and broader policies that address the structural determinants of food insecurity among people with TB [25]. Incorporating nutrition-sensitive approaches into TB treatment programs may enhance treatment success and improve the overall health outcomes among people with TB [2]. Such interventions are appropriate because undernutrition, for example, has been shown to reduce sputum smear conversion and treatment success among people with bacteriologically confirmed pulmonary TB [4], including increasing the risk of unsuccessful treatment outcomes [2]. Additionally, tackling underweight is crucial as malnourished people with TB are more likely to transmit TB to their household contacts [9]. Conversely, some studies have shown that overweight and mild obesity are paradoxically protective against mortality during TB treatment [26,27]. Although TB is a catabolic disease (causing wasting and weight loss), the studies argued that the extra energy stores might improve survival chances by reducing mortality. Another important implication concerns the link between obesity and diabetes mellitus. Obese people with TB who have diabetes mellitus have a higher risk of unsuccessful treatment outcomes [28], namely treatment failure, relapse, and death. Lastly, obese people with TB may experience worse lung function after TB cure, partly because obesity itself restricts lung expansion [29].

## Study strengths and limitations

The study strengths include a reasonably large sample size, hence enhances the reliability of the findings. Besides this, the key variables such as household food insecurity, dietary diversity, and alcohol consumption were assessed using standardized and validated instruments, hence strengthening internal validity. Unlike previous studies that focused on underweight or overweight/obesity alone, we examined both forms of malnutrition in a single study and identified the different determinants. The study limitations include unmeasured confounders such as food types and quantity, worm infestations, and medication side effects, among others, which could influence either underweight or overweight/obesity. Also, since this study was conducted in an urban setting, the findings may not fully generalize to rural populations due to differences in food access, dietary patterns, and socioeconomic conditions. Our study population included a high proportion of people with TB who also had HIV, which may have contributed to the high prevalence of underweight observed. Therefore, the findings may differ or may not perfectly generalize to settings where HIV prevalence among people with TB is lower. The findings may not also extrapolate to people with other forms of TB. Lastly, this was a cross-sectional study, so the findings demonstrate an association between the independent and outcome variables but not a causal relationship due to the absence of time sequence.

## Conclusions and recommendations

Our study showed that among people with clinically diagnosed and bacteriologically confirmed pulmonary TB, food insecurity is strongly associated with being underweight, likely due to insufficient access to nutritious foods and the metabolic effects of TB. Conversely, being overweight or obese is associated with being self-employed, likely due to improved access to food and reduced physical activity. Additionally, newly diagnosed individuals with TB are more likely to be overweight or obese than those with retreatment TB, probably due to the prolonged catabolic effects of TB disease and the impact of prior TB treatment. TB control programs, therefore, need to routinely assess nutritional status among people with TB, including integrating nutritional supplementation and social protection programs such as cash transfers and food aid. Moreover, broader policies that address the structural determinants of food insecurity may enhance treatment outcomes among people with TB.

## Supporting information

**S1 File.**
(CSV)

## Author contributions

**Conceptualization:** Simon Kyazze, Joseph Baruch Baluku, Jonathan Izudi.

**Data curation:** Simon Kyazze, Joseph Baruch Baluku, Jonathan Izudi.

**Formal analysis:** Saidi Appeli, Jonathan Izudi.

**Investigation:** Jonathan Izudi.

**Methodology:** Simon Kyazze, Saidi Appeli, Joseph Baruch Baluku.

**Project administration:** Simon Kyazze, Jonathan Izudi.

**Resources:** Simon Kyazze.

**Software:** Jonathan Izudi.

**Supervision:** Jonathan Izudi.

**Validation:** Saidi Appeli, Joseph Baruch Baluku, Jonathan Izudi.

**Visualization:** Saidi Appeli, Joseph Baruch Baluku, Jonathan Izudi.

**Writing – original draft:** Simon Kyazze, Saidi Appeli, Joseph Baruch Baluku, Jonathan Izudi.

**Writing – review & editing:** Simon Kyazze, Saidi Appeli, Joseph Baruch Baluku, Jonathan Izudi.

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
