## [Decision Letter · Decision Letter 0]

2 Dec 2025

PONE-D-25-22824Determinants of underweight and overweight/obesity among people with tuberculosis in Kampala, Uganda: a cross-sectional study.PLOS ONE

Dear Dr. Izudi,

Thank you for submitting your manuscript to PLOS ONE. After careful consideration, we feel that it has merit but does not fully meet PLOS ONE’s publication criteria as it currently stands. Therefore, we invite you to submit a revised version of the manuscript that addresses the points raised during the review process.

We look forward to receiving your revised manuscript.

Kind regards,

Paul Obeng, MEd, MSc., M.Phil.

Academic Editor

PLOS ONE

Journal Requirements:

3. Please amend your authorship list in your manuscript file to include author Joseph Baruch Baluku.

4. Please remove all personal information, ensure that the data shared are in accordance with participant consent, and re-upload a fully anonymized data set.

5. Please upload a copy of Supporting Information Figure S1 Fig, which you refer to in your text on page 19.

Reviewers' comments:

Reviewer's Responses to Questions

**Comments to the Author**

1. Is the manuscript technically sound, and do the data support the conclusions?

Reviewer #1: No

Reviewer #2: Yes

2. Has the statistical analysis been performed appropriately and rigorously? 

Reviewer #1: No

Reviewer #2: Yes

3. Have the authors made all data underlying the findings in their manuscript fully available?

Reviewer #1: No

Reviewer #2: Yes

4. Is the manuscript presented in an intelligible fashion and written in standard English?

Reviewer #1: No

Reviewer #2: Yes

5. Review Comments to the Author

Reviewer #1: Dear authors,

I congratulate you for taking time to work on the topic which actually is of global public importance. However, the submitted manuscript has the following serious errors:

1. The study population is not homogeneous (91.1% HIV cases vs 8.1% HIV negative cases). In my understanding the study population should be on HIV cases only OR the number of HIV negative subjects must be increased to almost equal or at least half of the HIV cases.

2. The multinomial regression was wrongly done because there is no specified reference group. in the methods authors stated that normal weight was their reference group but in Table 2 the reference group was again combined with underweight.

3. The percentages Table 1 were calculated in terms of column, this should be in row

General comment: Because of lack of homogeneity of the study population and wrong statistical analysis, in my views I think the work needs to be worked on comprehensively and resubmitted. Therefore, my recommendation is REJECT

Reviewer #2: Congratulations for developing this excellent paper! I believe that this is a welcome addition to the growing body of work on the relationship between nutrition and TB.

I have the following comments and suggestions:

1) Lines 102 to 105: Please mention the duration of data collection in this paragraph. I believe that you have mentioned it in line 131.

2) You have used various instruments to measure alcohol consumption, food insecurity and dietary diversity. Was any piloting done to test these instruments? If yes, please mention it in lines 138-144.

3) In lines 323-324, you have mentioned that malnourished people with TB are more likely to transmit TB to their household contacts. In addition, you can also mention that malnourished contacts are more likely to get infected with TB. This highlights the importance of nutrition supplementation, not just to people with TB, but also to their household contacts.

REF: Bhargava A, Bhargava M, Meher A, Benedetti A, Velayutham B, Sai Teja G, Watson B, Barik G, Pathak RR, Prasad R, Dayal R, Madhukeshwar AK, Chadha V, Pai M, Joshi R, Menzies D, Swaminathan S. Nutritional supplementation to prevent tuberculosis incidence in household contacts of patients with pulmonary tuberculosis in India (RATIONS): a field-based, open-label, cluster-randomised, controlled trial. Lancet. 2023;402:627-640. doi: 10.1016/S0140-6736(23)01231-X.

6. PLOS authors have the option to publish the peer review history of their article (what does this mean?). If published, this will include your full peer review and any attached files.

Reviewer #1: **Yes: **James Yahaya

Reviewer #2: No

---

## [Author Response · Author response to Decision Letter 1]

5 Dec 2025

Reviewer #1:

Dear authors, I congratulate you for taking time to work on the topic which actually is of global public importance. However, the submitted manuscript has the following serious errors:

1. The study population is not homogeneous (91.1% HIV cases vs 8.1% HIV negative cases). In my understanding the study population should be on HIV cases only OR the number of HIV negative subjects must be increased to almost equal or at least half of the HIV cases.

Response: Thank you for this insightful comment. Our study population comprises people with TB, regardless of HIV status. In Uganda, program data indicate that approximately 40% of people with TB have HIV, although this proportion varies across regions and sub-populations. TB remains one of the main opportunistic infections, and the leading cause of hospitalization and mortality among people with HIV. Conversely, HIV is the major driver of TB incidence. As such, the two epidemics are closely intertwined. That said, we acknowledge in the limitations section that our sample had a high proportion of people with TB who had HIV, which likely contributed to the high prevalence of underweight observed. We also note that findings may differ in settings where HIV prevalence among PWTB is lower. The new text on page 15 (lines 348-352) in the limitations section reads as follows:

“Our study population included a high proportion of people with TB who also had HIV, which may have contributed to the high prevalence of underweight observed. Therefore, the findings may differ or may not perfectly generalize to settings where HIV prevalence among people with TB is lower.”

2. The multinomial regression was wrongly done because there is no specified reference group. In the methods authors stated that normal weight was their reference group but in Table 2 the reference group was again combined with underweight.

Response: Thank you for this comment. Our outcome had three categories: underweight, normal weight, and overweight/obesity, with normal weight specified as the reference group, as described in the methods section. In Table 2, we clearly indicate this by labeling the comparison groups as “underweight vs. normal weight” and “overweight/obesity vs. normal weight.” Accordingly, the multinomial logistic regression was conducted using normal weight as the reference category when comparing individuals who were underweight or overweight/obese, which we believe is the appropriate approach.

3. The percentages Table 1 were calculated in terms of column, this should be in row

Response: Thank you for the comment. In cross-sectional studies with binary outcomes, row percentages are appropriate when the aim is to show how the outcome varies across levels of a covariate. However, when the goal is to describe how covariates are distributed within each outcome category, column percentages should be presented. When comparing groups, best practice is to use column percentages because the outcome is the column variable. Column percentages show the proportion of each covariate category within each outcome category, which is essential for interpreting group differences and aligns with how Chi-square comparisons are calculated and reported.

4. General comment: Because of lack of homogeneity of the study population and wrong statistical analysis, in my views I think the work needs to be worked on comprehensively and resubmitted. Therefore, my recommendation is REJECT

Response: We thank you for your overall assessment. We appreciate the concerns raised regarding population heterogeneity and the statistical analyses. As clarified in our responses to Comments 1-3, the study population appropriately reflects people with TB regardless of HIV status, which is consistent with national epidemiology and programmatic realities in Uganda. We have strengthened the limitations section to explicitly acknowledge how the distribution of HIV in the sample may have influenced the findings. Additionally, we have clarified the specification of the reference category in the multinomial logistic regression and the rationale for presenting column percentages in Table 1, both of which align with established analytic approaches. We believe that these clarifications, along with the corresponding revisions in the manuscript, adequately address the methodological concerns raised. We respectfully submit that the analyses were conducted appropriately and that the revised manuscript reflects these improvements.

Reviewer #2:

Congratulations for developing this excellent paper! I believe that this is a welcome addition to the growing body of work on the relationship between nutrition and TB. I have the following comments and suggestions:

1) Lines 102 to 105: Please mention the duration of data collection in this paragraph. I believe that you have mentioned it in line 131.

Response: We thank you for this comment. The data collection period (September 2, 2024, to November 29, 2024) is already reported under ‘Study variables and data collection’ (page 5, line 140). We suggest keeping it there for consistency, as moving it under ‘Study population and sampling’ would not substantially improve the manuscript’s clarity or integrity.

2) You have used various instruments to measure alcohol consumption, food insecurity and dietary diversity. Was any piloting done to test these instruments? If yes, please mention it in lines 138-144.

Response: Thank you for the comment. In our submission, we had already indicated that the data collection tools were pre-tested before the actual data collection. Under the section ‘Quality control measures’ on page 5 (lines 147-148), the text indicating the pre-testing of the instruments reads:

“To ensure data validity and reliability, the questionnaire was pre-tested in a non-study area before the data collection.”

3) In lines 323-324, you have mentioned that malnourished people with TB are more likely to transmit TB to their household contacts. In addition, you can also mention that malnourished contacts are more likely to get infected with TB. This highlights the importance of nutrition supplementation, not just to people with TB, but also to their household contacts.

REF: Bhargava A, Bhargava M, Meher A, Benedetti A, Velayutham B, Sai Teja G, Watson B, Barik G, Pathak RR, Prasad R, Dayal R, Madhukeshwar AK, Chadha V, Pai M, Joshi R, Menzies D, Swaminathan S. Nutritional supplementation to prevent tuberculosis incidence in household contacts of patients with pulmonary tuberculosis in India (RATIONS): a field-based, open-label, cluster-randomised, controlled trial. Lancet. 2023;402:627-640. doi: 10.1016/S0140-6736(23)01231-X.

Response: We are grateful for the sharing the useful resource. We agree with your suggestion. On page 3 (lines 79-81), we have added the text below, including the suggested reference by Bhargava et al (2023):

“Additionally, malnourished household contacts are more likely to get infected with TB than well-nourished household contacts. This highlights the importance of nutrition supplementation, not just to people with TB, but also to their household contacts.”

---

## [Decision Letter · Decision Letter 1]

27 Jan 2026

Determinants of underweight and overweight/obesity among people with tuberculosis in Kampala, Uganda: a cross-sectional study.

PONE-D-25-22824R1

Dear Dr. Izudi

We are pleased to inform you that your manuscript has been judged scientifically suitable for publication and will be formally accepted for publication once it meets all outstanding technical requirements.

Within one week, you will receive an e-mail detailing the required amendments. When these have been addressed, you’ll receive a formal acceptance letter, and your manuscript will be scheduled for publication.

Kind regards,

Paul Obeng, MEd, MSc., M.Phil.

Academic Editor

PLOS One

Reviewers' comments:

Reviewer #2: All comments have been addressed

---

## [Editor Report · Acceptance letter]

PONE-D-25-22824R1

PLOS One

Dear Dr. Izudi,

I'm pleased to inform you that your manuscript has been deemed suitable for publication in PLOS One. Congratulations! Your manuscript is now being handed over to our production team.

Kind regards,

on behalf of

Dr. Paul Obeng

Academic Editor

PLOS One